# Clinical Features and Management of Urethral Foreign Bodies in Children: A 10-Year Retrospective Study

**DOI:** 10.3390/children9101468

**Published:** 2022-09-26

**Authors:** Xiangpan Kong, Chun Wei

**Affiliations:** 1Department of Urology, Ministry of Education Key Laboratory of Child Development and Disorders, National Clinical Research Center for Child Health and Disorders, China International Science and Technology Cooperation Base of Child Development and Critical Disorders, Children’s Hospital of Chongqing Medical University, Chongqing 400014, China; 2Chongqing Key Laboratory of Pediatrics, Chongqing 400014, China

**Keywords:** emergency care, urethral foreign body, children, cystoscope, ultrasound

## Abstract

(1) Background: Urethral foreign bodies (UFBs) are very rare in children, and their treatment remains challenging. (2) Methods: A retrospective analysis was performed on 40 patients who were admitted to our hospital due to UFBs from June 2011 to June 2021. The clinical features and treatment experiences of these children are summarized. (3) Results: A total of 40 children were enrolled in the study, 17 boys and 23 girls. A majority of the boys (median age: 11.8 years) were of puberal age, and the main cause of the UFBs was sexual gratification (94.1%). Girls were almost always in early childhood (median age: 1.8 years), and most of the UFBs were related to specific clothing in specific regions and seasons. Ultrasound had a high accuracy in the diagnosis of female UFBs; the sensitivity and specificity were 88.9% and 85.7%, respectively. Most UFBs could be removed using a cystoscope (82.4% in boys, 100% in girls). All the children had a good prognosis and no complications occurred during follow-up. (4) Conclusions: Ultrasound is a reliable and sensitive method for the diagnosis of UFBs in girls. Cystoscopy is a reliable surgical method for UFBs.

## 1. Introduction

Urethral foreign bodies (UFBs) are very rare in children [1]. For boys, the relatively long and narrow urethra makes it difficult for exogenous foreign bodies to actively enter the urinary tract. Therefore, boys’ UFBs often occur in adolescence as a consequence of self-plugging for sexual satisfaction [2]. Once foreign bodies enter the urethra too deeply or enter the bladder, it is difficult to remove them by themselves, and they often require hospital treatment. For girls, due to their specific anatomical structure, the incidence of UFBs is much lower than that of vaginal foreign bodies [3,4,5] and is often not accompanied by obvious clinical symptoms (such as pain and hematuria), so there is a high risk of missed diagnosis.

Although the incidence of UFBs is low in children, their treatment remains challenging; for example, due to infection caused by improper removal of UFBs in boys; serious complications, such as fistula and even urethral stricture; and the risk of missed diagnosis in girls [6]. In addition, there is no consensus on treatment methods, with both open and minimally invasive approaches reported. Therefore, it is necessary to summarize the diagnosis and treatment experience of children with urethral foreign bodies. However, due to their low incidence, reviewing the existing literature, we found that almost all reports on children with urinary tract foreign bodies were published in the form of case reports, and there was still a lack of studies with large samples to summarize the clinical features and management experiences of children with UFBs. 

Hence, we reviewed the cases in our center in the last decade to determine the clinical features of urethral foreign bodies in children and summarize the experiences in management in order to raise awareness of the disease for early diagnosis and improve the outcomes of such rare conditions.

## 2. Materials and Methods

### 2.1. Data and Patients

A single-center retrospective analysis was performed on patients who were admitted to the Children’s Hospital of Chongqing Medical University due to UFBs from June 2011 to June 2021. This study was approved by the Ethics Committee of the Children’s Hospital of Chongqing Medical University (approval number: 2022-290, 19 June 2022). This retrospective study involving human participants was in accordance with the 1964 Declaration of Helsinki and its later amendments and comparable ethical standards. Written informed consent for their data to be used anonymously for statistical analysis was signed by the guardian of the child upon hospitalization.

The demographic data, chief complaint symptoms at the time of visit, imaging findings before surgery and conditions during surgery were analyzed.

### 2.2. Surgical Method

After general anesthesia, a rigid cystoscope or colposcope (STORZ^®^(KARL STORZ Endoscopy (Shanghai, China) Ltd., KARL STORZ China Center. Shanghai, China), Fr9.5, 0°) was used as a routine means of examination and treatment, and open surgery was performed according to the specific situation for special types of UFBs that are difficult to remove under a cystoscope (Figure 1).

Special type of UFBs: electric wires, which cannot be removed under cystoscopy.

### 2.3. Postoperative Management

Antibiotics were administered based on whether the child had a urinary tract infection. In children who required intravenous treatment, tobramycin or gentamicin was used and normal kidney function was confirmed. When abnormal kidney function was suspected, ceftriaxone or cefotaxime were alternative treatment options. In children who can receive oral treatment without any known resistant urinary cultures, cefixime or amoxicillin-clavulanate are the empirical treatment options. The selection of antibiotics was flexible according to the actual situation [7]. Whether to catheterize or not and the duration of catheter indwelling should be related to the degree of urethral injury and the presence or absence of a UTI. For children with little or no urethral injury, no catheterization or temporary catheterization after surgery is safe; however, for those who have injured the urethra with a foreign object, a short catheterization (within 3 days) can reduce bleeding and relieve pain. Finally, for those children with a definite or high suspicion of having contracted a complicated UTI, catheterization for a week or so is necessary [8,9].

Children returned to the outpatient department for review once within 1 month after surgery. In addition to asking about clinical symptom relief, a routine urinalysis and ultrasonography were usually performed to evaluate the outcome of the surgery.

### 2.4. Statistical Analysis

All patient information is descriptively presented as averages and standard deviations. Formal analysis was performed.

## 3. Results

### General Data

A total of 40 patients were admitted to our hospital (Figure 2), aged from 10 month to 15 years with a mean age of 6.6 ± 5.1 years. Boys were aged from 4.5 to 15 years, with a mean age of 11.7 ± 2.5 years. Girls were aged from 10 mo to 12.5 years, with a mean age of 2.9 ± 2.8 years (Figure 3). All 17 boys presented with pain (nine cases), pain with hematuria (four cases), hematuria (three cases), or pain with infection (one case). The duration of symptoms ranged from 2 h to 6 mo. Among the 23 girls, 3 were treated for pain and 2 for hematuria. The remaining girls had no obvious symptoms, since they were conducted to the hospital after their parents had found UFBs entering the urethral meatus that could not be removed. The duration of symptoms ranged from 4 h to 2 months. No foreign bodies were found in 13 girls during surgery, 1 child’s parents refused surgery, and the other 9 children had foreign bodies. The types of UFBs in these girls were a paddy (six cases), rubber hose (two cases) and paper clip (one case), and in no cases were two or more simultaneous different types of foreign bodies ever found (Figure 4). The foreign body types in boys were a needle (six cases), magnetic beads (three cases: 36, 18 and 52), bamboo stick (three cases), electric wires (two cases), rubber hose (two cases) and surgical suture (one case). All UFBs were removed successfully under cystoscope, except for the first case of magnetic beads in our hospital, the case of surgical sutures and the two cases of electric wires.

Preoperative X-ray examination was performed on all boys to determine the location and size of the foreign bodies. All 23 girls underwent preoperative ultrasound. Among them, the ultrasound reports of 12 girls were negative, and one of these 12 girls did not undergo surgery because their parents refused it. The surgical results of the remaining 11 girls were negative, and no foreign bodies were found, which was consistent with the ultrasound results. The other 11 girls with positive ultrasound findings received surgical treatment, and nine cases of foreign bodies were found, all of which were successfully removed, while no foreign bodies were found in the other two cases. It is worth noting that the parents of one of these two children clearly reported a history of self-removal of foreign bodies after preoperative ultrasound examination (Table 1). The average indwelling duration of the catheter was 3.5 ± 1.3 day and 0.5 ± 1.0 d for boys and girls, respectively.

Postoperative recovery was satisfactory. The follow-up period was 3 months to 2 years. No perforation or fistula was found, and boys had no urethral stricture.

## 4. Discussion

The occurrence of UFBs in children is very rare, but it is challenging. Our study provided a large sample size, and we summarized the clinical features and management experiences of UFBs to improve the management of such rare conditions.

On the age and the cause, boys and girls had distinct characteristics; for boys, most occurred in adolescence, mostly out of autologous sexual stimulation, and most of these UFBs could not be removed by the patient themselves after insertion. Metal needles or tubes may stay in the urethra, causing pain or hematuria in children, while some other UFBs, such as magnetic beads, can cause pain, urinary tract infection and other discomfort after falling completely into the bladder. In our study, we found that early UFBs were mainly metal needles, sticks and wires, while in recent years, the incidence of magnetic beads has been increasing year by year, which may be related to incomplete sex education and unhealthy influences brought about by the use of social networks [10].

In girls, on the other hand, UFBs tend to occur at a very young age, caused by an accident. In the summer, children in China are often dressed in split pants; as a result, children dressed in split pants playing in drying paddies may result in UFBs occurring. This is consistent with literature reports of accidental damage to the urinary organs happening during the summer [11]. These children often have no obvious clinical symptoms and are therefore more likely to be missed. Therefore, we recommend that younger children not play with toys with small accessories to avoid similar situations. Fortunately, in recent years, with the improvement of hygiene practices, the habit of children wearing split pants has gradually changed, which has greatly reduced the occurrence of such accidents.

The particularity of children makes it very difficult to collect their medical history, and diagnosis is difficult both for young children, who have difficulty expressing themselves, and for older children, who may hide medical problems out of shame [12]. Therefore, it is necessary to select a reliable and accurate examination method. For boys, pelvic X-ray is effective for most foreign bodies [13,14]. It can effectively provide the size, shape and location of UFBs and provide guidance for surgery. For girls, our advice is to use ultrasound for diagnosis [15]. In our study, preoperative ultrasonic examination results and intraoperative findings were almost completely consistent, with only one case without surgery and two cases where nothing was found intraoperatively despite a positive preoperative ultrasound. In view of the fact that two children were observed to have clear foreign body expulsion during their hospitalization before surgery, we suspected that the foreign bodies in these two children might have been discharged by themselves after examination. It is noteworthy that there may be some deviation in the positioning of the foreign bodies under ultrasound. In this study, we found that in two cases who underwent preoperative ultrasound, removal of a vaginal foreign body was confirmed during intraoperative exploration of a bladder foreign body, which was related to the physiological anatomy; therefore, for girls, our experience is that, after a very close examination of a vaginal foreign body, a urinary tract exploration is necessary and vice versa.

Cystoscopy/colposcopy is an ideal treatment for UFBs. In this study, UFBs were successfully removed under cystoscopy, except for one case of coiled wires in the bladder, the first case of magnetic beads and one case of stones caused by surgical sutures accidentally sewn into the bladder after oblique hernia.

Different from the female urethra, the male urethra is narrower and physiological bending often occurs, making male UFBs more difficult to remove, especially for UFBs in the bladder. Magnetic beads, for example, once inside the bladder, are magnetically attracted to each other, making them very difficult to remove [16]; as a result, many doctors will choose an open or laparoscopic approach to remove magnetic beads [17,18,19,20]. In our first case of such children, we also adopted an open method to remove the magnetic beads, but in the subsequent cases, we adopted the method of foreign body forceps under cystoscope to successfully remove bladder magnetic beads in two cases. Our experience involved two surgeons collaborating on the operation, using foreign body forceps on the edge at first to grab the beads, drag the clamped magnetic bead into the urethra and slowly drag it back and forth between the bladder and urethra several times in order to attract the first magnetic bead to the rest of the magnetic beads and arrange them in series in the urethra to extract more magnetic beads simultaneously. For the first few times, it was only possible to pull out fewer magnetic beads, maybe because of the large clouds of magnetic beads inside the bladder; however, with the decrease in the number of magnetic beads in the bladder, the number of magnetic beads removed increased significantly each time. In the most recent case, we successfully removed 52 whole magnetic beads, and a maximum of 22 beads were removed at one time. Therefore, cystoscopic removal of beads is feasible and efficient with certain skills. Similar success has been reported by Ellimoottil et al. [21].

Considering the unreliability of the history, the limitations of auxiliary examination and the limitations of endoscopic probes, it is necessary to carry out relevant checks again before the end of surgery, especially for some special types of foreign bodies. For example, in our recent processed cases, before surgery, a boy told us he had placed about 30 magnetic beads, and preoperative pelvic radiographs could not clearly show the number of beads. However, a total of 52 beads were removed intraoperatively (Figure 5). Therefore, intraoperative ultrasound and bedside radiographs were performed to confirm that all foreign bodies were removed before we decided to end the surgery.

Admittedly, there are some limitations to the study, such as its retrospective nature and data from a single center. Therefore, the conclusions of this study—that boys and girls have completely different ages and causes of onset, that ultrasound is a reliable method to diagnose UFBs and that cystoscopy is a reliable surgical method to treat UFBs—still need to be further verified by other prospective studies with larger samples.

## 5. Conclusions

The occurrence of UFBs in children is very rare, but it is challenging. Timely and effective management can improve outcomes for these children.

Boys and girls have completely different clinical features. Ultrasound is a reliable method to diagnose UFBs, and cystoscopy is a reliable surgical method to treat them.

We hope that our study can provide diagnosis and treatment experiences for such rare conditions, improve the outcomes of such conditions, and improve treatments for these children.

## Figures and Tables

**Figure 1 children-09-01468-f001:**
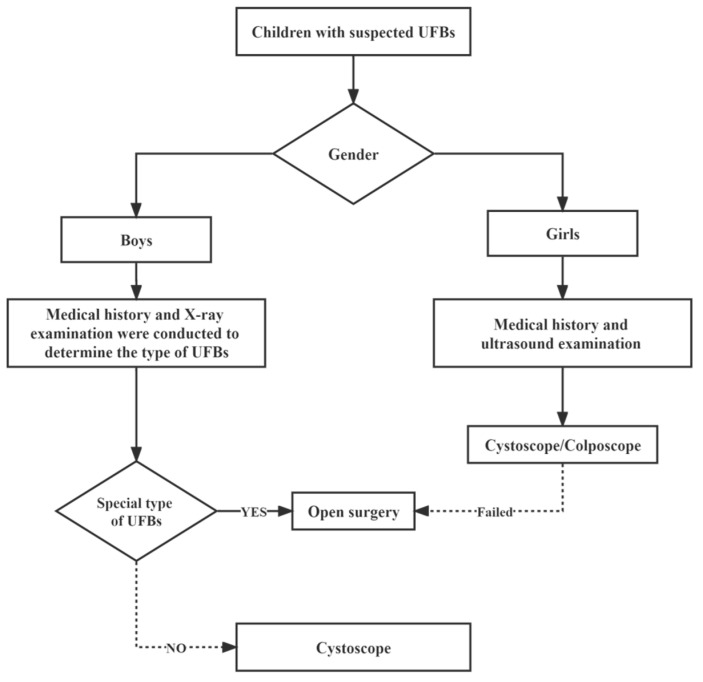
Management of children with UFBs.

**Figure 2 children-09-01468-f002:**
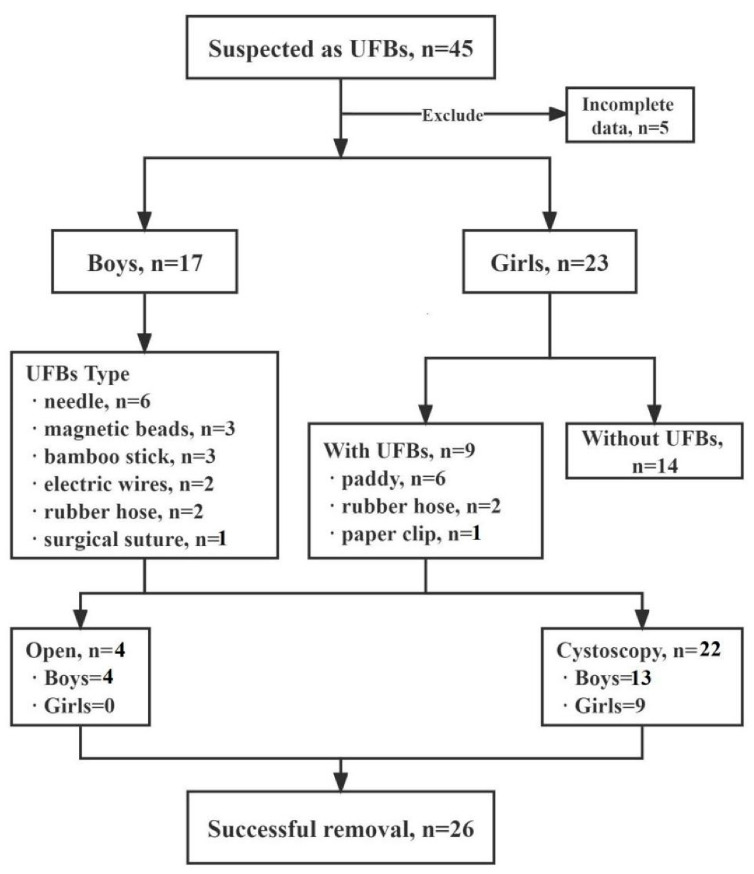
Flow chart of the study.

**Figure 3 children-09-01468-f003:**
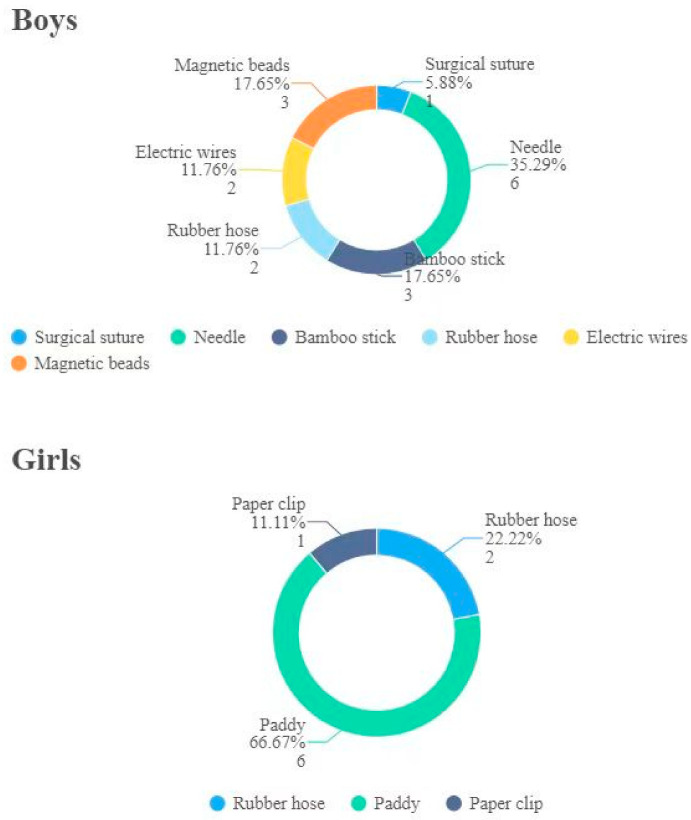
Type and quantity of urethral foreign bodies in boys and girls.

**Figure 4 children-09-01468-f004:**
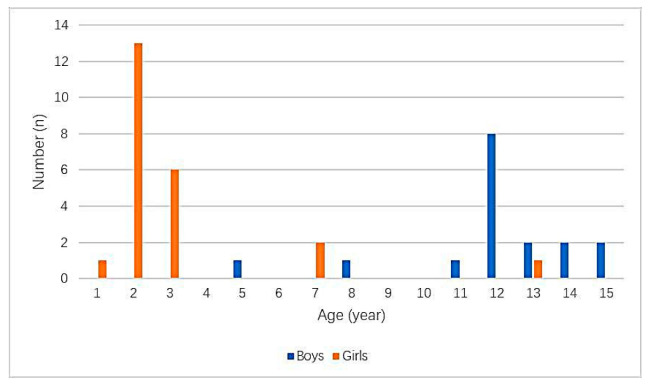
Age distribution of urethral foreign bodies in children.

**Figure 5 children-09-01468-f005:**
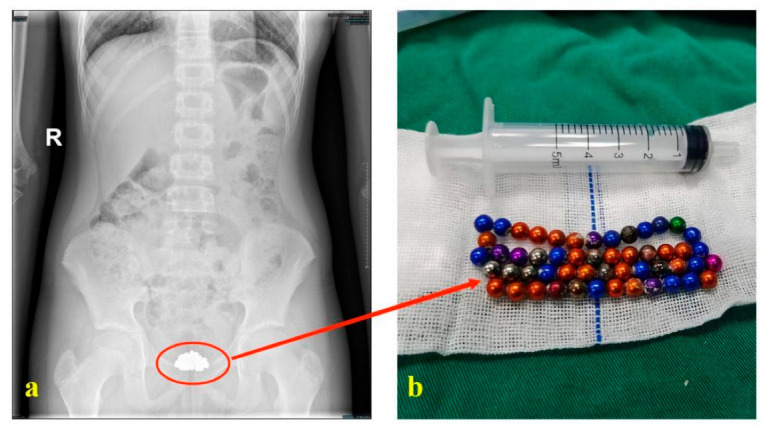
Preoperative examination and postoperative physical picture of magnetic beads. ((**a**): Preoperative X-ray examination of the patient; (**b**): Magnetic beads removed after surgery).

**Table 1 children-09-01468-t001:** Characteristics of the UFBs and their management.

Case	Age (Years)	Gender	Clinical Symptoms	Duration	X-ray	Ultrasound	Surgical Method	UFB Location	Type of UFB
1	11.6	M	Pain and bleeding	6 h	+	ND	Cystoscope	Urethra	Needle
2	8.0	M	Pain	4 h	+	ND	Cystoscope	Urethra	Needle
3	0.8	F	No special symptoms	48 h	ND	-	Cystoscope	None	None
4	13.9	M	Pain and infection	48 h	+	ND	Cystoscope	Urethra	Bamboo stick
5	1.7	F	No special symptoms	24 h	ND	-	Cystoscope	None	None
6	11.6	M	Pain and bleeding	26 h	+	ND	Cystoscope	Urethra	Needle
7	2.3	F	Pain	96 h	ND	-	Cystoscope	None	None
8	1.6	F	No special symptoms	24 h	ND	-	Cystoscope	None	None
9	1.6	F	No special symptoms	4 h	ND	-	Cystoscope	None	None
10	2.9	F	No special symptoms	12 h	ND	-	Refused	None	None
11	3.0	F	No special symptoms	22 h	ND	-	Cystoscope	None	None
12	1.2	F	No special symptoms	24 h	ND	-	Cystoscope	None	None
13	12.0	M	Pain	24 h	+	ND	Cystoscope	Urethra	Needle
14	12.8	M	Pain and bleeding	2 h	+	ND	Cystoscope	Urethra	Rubber hose
15	14.7	M	Pain and bleeding	8 h	+	ND	Cystoscope	Urethra	Bamboo stick
16	11.2	M	Pain	20 h	+	ND	Cystoscope	Urethra	Needle
17	8.3	F	Bleeding	2 mo	ND	+	Cystoscope	Urethra and bladder	Rubber hose
18	11.0	M	Pain	8 h	+	ND	Cystoscope	Urethra	Needle
19	3.0	F	Bleeding	24 h	ND	-	Cystoscope	None	None
20	1.6	F	No special symptoms	24 h	ND	-	Cystoscope	None	None
21	1.9	F	No special symptoms	24 h	ND	+	Cystoscope	Bladder	Paddy
22	15.0	M	Pain	2 mo	+	ND	Cystoscope	Urethra and bladder	Rubber hose
23	13.9	M	Bleeding	1 mo	+	ND	Cystoscope	Urethra and bladder	Rubber hose
24	11.8	M	Bleeding	24 h	+	ND	Open	Urethra and bladder	Electric wires
25	2.2	F	No special symptoms	12 h	ND	+	Cystoscope	Bladder	Paddy
26	2.0	F	No special symptoms	24 h	ND	+	Cystoscope	None	None
27	1.4	F	No special symptoms	14 h	ND	+	Cystoscope	Bladder	Paddy
28	1.8	F	No special symptoms	24 h	ND	+	Cystoscope	Bladder	Paddy
29	12.8	M	Pain	14 h	+	ND	Open	Bladder	Magnetic beads (36)
30	1.8	F	No special symptoms	24 h	ND	+	Cystoscope	None	None
31	1.8	F	No special symptoms	7 h	ND	+	Cystoscope	Bladder	Paddy
32	11.3	M	Pain	6 h	+	ND	Cystoscope	Bladder	Magnetic beads (18)
33	8.3	F	Pain	24 h	ND	+	Cystoscope	Urethra and bladder	Rubber hose
34	12.5	F	Pain	10 h	+	+	Cystoscope	Urethra	Paper clip
35	12.0	M	Pain	5 day	+	ND	Cystoscope	Bladder	Magnetic beads (52)
36	1.7	F	No special symptoms	3 day	ND	-	Cystoscope	None	None
37	1.3	F	No special symptoms	24 h	ND	+	Cystoscope	Bladder	Paddy
38	2.2	F	No special symptoms	24 h	ND	-	Cystoscope	None	None
39	10.8	M	Bleeding	40 day	+	ND	Open	Urethra and bladder	Electric wires
40	4.5	M	Pain	6 mo	+	ND	Open	Bladder	Surgical suture

ND: Not found, M: male, F: female.

## Data Availability

Not applicable.

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
