# Peer review of "Clinical Features and Management of Urethral Foreign Bodies in Children: A 10-Year Retrospective Study"

_children, 2022, doi:10.3390/children9101468_

Round 1

Reviewer 1 Report

First, I would like to congratulate the authors on their work. This retrospective study shows a single-center experience of the management of pediatric urethral foreign bodies. 

The authors have shown their results descriptively in 40 children who were managed in the last decade. There is nothing new in this manuscript. I have several comments. 

Introduction: What was your hypothesis before conducting this study? Please add your hypothesis in 2-3 lines at the end of the Introduction section.

Methods: What is your protocol of management of UFB? Please give the algorithmic management based on: symptomatic/asymptomatic, boy/girl, witnessed/unwitnessed, etc.

-How did you analyze the data? Was it only descriptively presented? If yes, please add in the methods section that- All patient information is descriptively presented as averages and standard deviations. No formal analysis was done.

Results:

-What is the name of Subsection 3.1? Please add it.

-What do you mean by course of the disease? Is it the duration of symptoms? Please be specific as it is not a chronic disease but an acute event.

-In Figure 1, in the boys' section, the n comes out to be 18 (after totaling) and not 17. is it a typographical error or do some cases have more than one foreign body?

-Please add legends to Table 1.

-Figures 2 and 3 are unclear. Please provide high-resolution images.

-What is meant by "first case of magnetic beads"? You have used it many times in the manuscript. Please simplify the term.

There are major grammatical errors throughout the manuscript. A writing assistant is strongly needed as some of the sentences are totally non-comprehensible. For example:

--despite 1 cases without surgery and 2 cases underwent preoperative ultrasonic prompt positive and intraoperative no found, in view of the fact that 2 children were observed to have clear foreign body expulsion during their hospitalization before surgery, we suspected that the foreign bodies in these two children might have been discharged by them- 145 selves after examination, but we did not found it

-Noteworthy, ultrasonic to the positioning of the foreign bodies maybe there is some deviation

Author Response

Dear Reviewer:

Thank your comments concerning our manuscript "ID 1865825", titled "Clinical features and management of urethral foreign bodies in children:a 10-year retrospective study" Those comments are all valuable and very helpful for revising and improving our paper, as well as the important guiding significance to our researches. We have studied comments carefully and have made correction which we hope meet with approval. Revised portion are marked red in the paper. The corrections in the paper and the responds to the comments are as flowing:

Generally speaking: We answered all comments one by one. We revised the manuscript according to suggestion of reviewers in main text, figures, tables and central image. We asked an English native language expert in the area of urology and four famous surgeon in the area to help as revised the manuscript. We changed some descriptions to make our manuscript more accurate and precise. We do hope these efforts can promote the quality of the manuscript.

Comment 1: There are major grammatical errors throughout the manuscript. A writing assistant is strongly needed as some of the sentences are totally non-comprehensible.

Answer 1: As a non-native English speaker, we have made many mistakes in the process of writing this manuscript, we are so sorry for that, and we have invited an English native language expert to check the revised version of our paper to ensure correctness of the spelling, grammar and syntax. We hope our efforts can improve the language quality of the manuscript. The detailed modifications are listed below. Thank you very much for your careful work. We know it took your precious time.

Change 1: We have invited an English native language expert to check the revised version of our paper to ensure correctness of the spelling, grammar and syntax.

Introduction

  • What was your hypothesis before conducting this study? Please add your hypothesis in 2-3 lines at the end of the Introduction section.

Answer: Thanks for your question. We are so sorry for this, and we added this part of content in the revised manuscript.

Change: We added the description in the Introduction section " Hence, we reviewed the cases in our center in the last decade, to find out the clinical features of urethral foreign bodies in children, summarize the experiences in managing, to raise awareness of the disease for early diagnosis and improve the outcomes of such rare diseases." in line 48-51, page 2.

Methods

  • What is your protocol of management of UFB? Please give the algorithmic management based on: symptomatic/asymptomatic, boy/girl, witnessed/unwitnessed, etc.

Answer: Thanks for your suggestion. For this paper, it is necessary to clearly describe our protocol of management of UFBs, so we draw Figure 1 to show it. 

Change: We added the new Figure 1 to describe our protocol of management of UFBs.

  • How did you analyze the data? Was it only descriptively presented? If yes, please add in the methods section that- All patient information is descriptively presented as averages and standard deviations. No formal analysis was done.

Answer: We fell so sorry that we did not describe the statistical methods, and we added it in the revised manuscript.

Change: We added “All patient information is descriptively presented as averages and standard deviations. No formal analysis was done.” in the methods section. (Line81-82, Page 3)

Results:

  • What is the name of Subsection 3.1? Please add it.

Answer: Thank you for your careful review, we missed that, and we added it in the revised manuscript.

Change: We added “General data” in Line 84, Page 3.

  • What do you mean by course of the disease? Is it the duration of symptoms? Please be specific as it is not a chronic disease but an acute event.

Answer: Thank you for your professional advice, “the duration of symptoms” is much more appropriate and professional.

Change: We have corrected “course of the disease” to “the duration of symptoms” in the revised manuscript. (Line93, Page3)

  • In Figure 1, in the boys' section, the n comes out to be 18 (after totaling) and not 17. is it a typographical error or do some cases have more than one foreign body?

Answer: It is a typographical error, we have corrected the mistake.

Change: We have corrected the mistake and uploaded the figure in the revised manuscript.

  • Please add legends to Table 1.

Answer: Thank you for your careful review, we missed that, and we added it in the revised manuscript.

Change: We added “Table 1. Characteristics of the UFBs and their management.” in the revised manuscript. (Line116 , Page4)

  • Figures 2 and 3 are unclear. Please provide high-resolution images.

Answer: We are so sorry about that, and we provided high-resolution images in the revised manuscript. .

Change: We provided high-resolution images in the revised manuscript.

  • What is meant by "first case of magnetic beads"? You have used it many times in the manuscript. Please simplify the term.

Answer: We mean “the first case of magnetic beads in our hospital” , we have corrected the wrong phrase.

Change: We have corrected the wrong phrase. (Line101 , Page3)

Best Regards,

Xiangpan Kong, M.Med.

Reviewer 2 Report

How do you explain the fact that UFB in girls is not accompanied by obvious clinical symptoms?

Do you have data on the numerical incidence of UFB in relation to gender and age (generally)?

The introduction needs to be revised more extensively with a more detailed description of possible complications and treatment methods.

In accordance with legal regulations, from whom did you receive informed consent? Please state the same clearly in your manuscript.

Please explain exactly what are the indications for open surgery and what are the possible complications compared to cystoscopy.

According to the guidelines, what are the recommended antibiotics for infection control during urinary tract injury? Please provide a reference.

According to the guidelines, until how many days is catheterization of children recommended in relation to age and gender? Are there complications of long-term catheterization? Please provide a reference.

The results section is not complete! A number of sentences remained from the original template.

The numerical data in Figure 1 do not match.

According to your results, how do you explain the fact of higher incidence in girls at an earlier age?

You state that an open surgery or cystoscopy was performed during which no foreign bodies were found, and where the same were not observed by diagnostic methods? How do you explain that?

What diagnostic methods, apart from X-ray and ultrasound, could you have used?

The manuscript must be proofread by a native English speaker!

After reading the title, I expected much more from the manuscript. The manuscript needs to be thoroughly revised from the beginning and possibly re-sent for consideration.

Author Response

Dear Reviewer:

Thank your comments concerning our manuscript "ID 1865825", titled "Clinical features and management of urethral foreign bodies in children:a 10-year retrospective study" . We have carefully taken your comments into consideration in preparing our revision, which has resulted in a paper that is clearer and more compelling. Revised portion are marked red in the paper. The corrections in the paper and the responds to the comments are as flowing:

Generally speaking: We answered all comments one by one. We revised the manuscript according to suggestion of reviewers in main text, figures, tables and central image. We asked an English native language expert in the area of urology and four famous surgeon in the area to help as revised the manuscript. We changed some descriptions to make our manuscript more accurate and precise. We do hope these efforts can promote the quality of the manuscript.

Comment 1: How do you explain the fact that UFB in girls is not accompanied by obvious clinical symptoms?

Answer 1: Thank you for your questions. I think there are two reasons to explain this phenomenon. Firstly, the girls in this group are very young(about two years old on average), they are too young to show any obvious clinical symptoms. Secondly, the type of UFB may have something to do with it, we checked the data and found that the UFBs in these children were mostly paddy, paddy entering the urethra does not cause the same severe irritation as other UFBs.

Change 1: We added “These children often have no obvious clinical symptoms and are therefore more likely to be missed.” to the discussion section to explain this result and to remind you that this is also a point of clinical attention because it may lead to missed diagnosis.(Line153-154, Page8)

Comment 2:Do you have data on the numerical incidence of UFB in relation to gender and age (generally)?

Answer 2: Thank you for your questions. As we introduced in the article, children's UFB are a rare disease with a very low incidence. Our hospital is the largest children's medical center in southwest China, and only 40 cases of UFB have been treated in the past ten years. In addition, the current literature on urethral foreign bodies is mostly a case report. Therefore, we do not have data on the incidence of UFB in relation to gender and age.

Comment 3: The introduction needs to be revised more extensively with a more detailed description of possible complications and treatment methods.

Answer 3: Thank you for your suggestion. With the help of you and other reviewers, we almost rewrote the introduction section, which made our manuscript more accurate and precise. We added a description of complications and treatment methods.

Change 3: We added “Although the incidence of UFBs is low in children, its treatment remains challengingdue to, for example, infection caused by improper removal of UFBs in boys, serious complications such as fistula and even urethral stricture, and the risk of missed diagnosis in girls[6]. In addition, there is no consensus on treatment methods, with both open and minimally invasive approaches reported. ” to the introduction section.(Line38-42, Page1)

Comment 4: In accordance with legal regulations, from whom did you receive informed consent? Please state the same clearly in your manuscript.

Answer 4: Thank you for your suggestion. As required by law, we obtained informed consent from the parents or guardians of the children when they were admitted to the hospital.

Change 4: We add “Written informed consent for their data to be used anonymously for statistical analysis was signed by the guardian of the child when hospitalized.” to the  Materials and Methods section.(Line60-61, Page2)

Comment 5: Please explain exactly what are the indications for open surgery and what are the possible complications compared to cystoscopy.

Answer 5: Thank you for your suggestion. As one of the most important points in this study, it is necessary to explain exactly the indications for open and cystoscopic surgery, The indications for the two procedures are described in detail in Figure 1 in the Methods section. Compared with cystoscopic surgery, the biggest disadvantage of open surgery is greater trauma, and it has been reported that long-term urethral stricture is caused by incision of male urethra.

Change 5: We added the new “Figure 1” to the Materials and Methods section.

Comment 6:According to the guidelines, what are the recommended antibiotics for infection control during urinary tract injury? Please provide a reference.

Answer 6:Thank you for your suggestion. According to EAU guidelines, The choice of agent is also based on local antimicrobial sensitivity patterns, and should later be adjusted according to sensitivity-testing of the isolated uropathogen. In children who require intravenuous treatment tobramycin or gentamicin is recommended if there is normal kidney function. When abnormal kidney function is suspected, ceftriaxon or cefotaxime are alternative treatment options. In children who can receive oral treatment without any known resistant urinary cultures, cefixime or amoxicillin-clavulanate are the empirical treatment options. The selection of antibiotics should be flexible according to the actual situation. We cite the recently published literature on the standard use of antimicrobial agents.

Change 6:We added “[7]” , the “Update of the EAU/ESPU guidelines on urinary tract infections in children. PMID:33589366 ” to introduce the selection rules of antibiotics in detail.

Comment 7: According to the guidelines, until how many days is catheterization of children recommended in relation to age and gender? Are there complications of long-term catheterization? Please provide a reference.

Answer 7: Thank you for your questions.

(1)We believe that whether to catheterize or not and the duration of catheter indwelling should be related to the degree of urethral injury and the presence or absence of UTI, For children with little or no urethral injury, no catheterization or temporary catheterization after surgery is safe, for those who have injured the urethra with a foreign object, a short catheterization (within 3 days) can reduce bleeding and relieve pain, finally, for those children with definite or high suspicion of complicated UTI, catheterization for a week or so is necessary.

(2) For children, the main possible complications of long-term catheterization are secondary urinary tract infections, catheter blockage, and catheter slip.

Change 7: We added two references([8-9]) describing the choice of catheter indwelling duration and complications of long-term catheterization respectively.

Comment 8: The results section is not complete! A number of sentences remained from the original template.

Answer 8: Thank you for your suggestion. In the revised version, we rewrote the results section and removed the sentences from the original template.

Change 8: We rewrote the results section and removed the sentences from the original template.

Comment 9: The numerical data in Figure 1 do not match.

Answer 9: Thank you for your suggestion. It is a typographical error, we have corrected the mistake.

Change 9: We have corrected the mistake and uploaded the figure in the revised manuscript.

Comment 10: According to your results, how do you explain the fact of higher incidence in girls at an earlier age?

Answer 10: Thank you for your question. As we wrote in the article “In the summer, children in our country are often dressed in split pants; as a result, children dressed in split pants playing in drying paddies may cause UFBs to occur. This is consistent with literature reports of accidental damage to the urinary organs happening during the summer.”(Line150-153, Page8).

 I wonder if you agree with this speculation?

Comment 11: You state that an open surgery or cystoscopy was performed during which no foreign bodies were found, and where the same were not observed by diagnostic methods? How do you explain that?

Answer 11: Thank you for your question. On this point, we also made a reasonable speculation and explanation in the article.

“In our study, preoperative ultrasonic examination results and intraoperative findings were almost completely consistent, with only one case without surgery and two cases where nothing was found intraoperatively despite a positive preoperative ultrasound. In view of the fact that two children were observed to have clear foreign body expulsion during their hospitalization before surgery, we suspected that the foreign bodies in these two children might have been discharged by themselves after examination.”(Line165-171, Page8)

We don't know if such a conjecture can get your professional approval.

Comment 12: What diagnostic methods, apart from X-ray and ultrasound, could you have used?

Answer 12: Thank you for your question. apart from X-ray and ultrasound, we didn't use many other diagnostic methods, actually, cystoscopy is also an important method for diagnosis.

Comment 13: The manuscript must be proofread by a native English speaker!

Answer 13: Thank you for your suggestion. We are so sorry for that, and we have invited an English native language expert to check the revised version of our paper to ensure correctness of the spelling, grammar and syntax. We hope our efforts can improve the language quality of the manuscript. The detailed modifications are listed below. Thank you very much for your careful work. We know it took your precious time.

Change 13: We have invited an English native language expert to check the revised version of our paper to ensure correctness of the spelling, grammar and syntax.

Best Regards,

Xiangpan Kong, M.Med.

Reviewer 3 Report

The Authors show their large experience in the treatment of urethral foreign bodies in children. This is a single-center retrospective study.

The design is clear, methodology is correct, figures are appropriate. References are updated.

Dear Authors,

I am glad to review your interesting work. 

Nevertheless, I would like to explain my suggestions to improve the quality and the scientificity of your manuscript. 

Minor english spell check is required. 

If you will not accept any point, I eventually ask you to gently explain the reasons. 

ABSTRACT

The text is well articulated, minor text editing are suggested. 

Line 16: a space missing before "its treatment"

Line 19: I suggest to delete "eventually"  

Line 20: the repetition of "most-mostly" makes the text rough. You can reformulate for example: the majority of boys was in puberal age and the main cause was the sexual gratification. 

Line 21: here you can separate the two sentences (i.e. ". Girls are almost"). Please, avoid the too long sentences. 

Line 23: "and seasons (when grain is mature in rural areas)" is a deepening that does not allow an understanding of the issue. You may delete it or reformulate. 

Line 23-25: the phrase is too long and it is difficult to understand. Please improve also english syntax. 

Point 1: you may specify if you refer to a rigid cystoscope or a flexible.

INTRODUCTION

Line 36: "therefore, boy's UFBs often occur in adolescence" is a correct logic consequence of the previous phrase, but I suggest to add "as a consequence of self-plugging for sexual satisfaction", since this information is actually missing in the introduction. 

Line 40: "obvious clinical symptoms". I completely agree with you that related symptoms are obvious, but I suggest to specify them (since they are not in the entire introduction section) to allow the text understanding also to the readers who are not surgeons/urologists (according to Journal Scope). 

MATERIALS AND METHODS

Some contents are missing. 

2.1 Data and patients

Point 2: in this section, as correctly stated in the abstract, authors should say "retrospective analysis was performed on patients who were admitted to our hospital due to UFBs from June, 2011 to June, 2021.". Add also this informations: a single-center and the name of the hospital center.

Line 60: figure 1 contents are presented in the results section. I suggest to move it in the results section, since It shows some results.

2.2 Surgical method

Line 62: please specify rigid or flexible cystoscope, which brand, degrees of optics and caliber.

Point 3: all the endoscopic procedures were performed under general anestesia? If you could resume this data you would add it. 

Line 66: "indwelling catheterization for several time". It would be interesting if the authors specified the average time or standard deviation of catheterization in the results section. 

Line 68: what do the authors intend for "review"? Is is a re-cystoscopy?

RESULTS

According with MDPI Guidelines, it would not exist a section only for figures, tables and schemes, since they have to be integrated in the main text. 

I suggest to delete line 71-74.

Line 76: since the UFBs medium age is totally different in each gender (as the authors demonstrated), I suggest to offer a median age and SD calculated:

- for the male group

- for the female group

- for the entire sample of patients

The text may be rearranged as follows: a total of X patients wad admitted to our hospital, aged from .... to ...., with a mean age .... (SD ...). Boys were aged ...., mean .... . Girls were ...., mean..... . This construction, would enhance the size of the sample and would offer some prompt data for the eventual future use of your work.

This information can be also added into the figure 2, to have a "pic" of the sample. 

Line 77: "with or without"? Please specify how many of them had hematuria beyond the pain. 

Line 79: "the other children" is "the remaining girls"? 

Line 79-80: "mainly because...." can be improved. For example you can say "since they were conducted to the hospital after their parents had found UFBs entering the urethral meatus that could not be removed". 

Line 87: a further conclusion would be "No simultaneous two or more different types of foreign bodies have been ever found".

Line 88-95: since the diagnostic assessment usually precedes the management, this section might be replaced before "the foreign body types (line 81-82).

Line 96: "stable" is "uneventful, regular"? Please, choose a more appropriate term. 

Line 96: "the follow-up period". How did they performed follow up? Uroflowmetry? Micturitction diary? Cystoscopy? This information is missing and needs to be specified in the section "materials and methods".

Line 97: an exceeding point at the end.

FIGURES

Please, ensure an appropriate quality according with 7.1 Figures - MDPI Guidelines https://www.mdpi.com/authors/layout#_bookmark39 

Actually, for example, figure 3 is hardly readable. 

DISCUSSION

The contents are appropriate, nevertheless:

- the authors start focusing the section on their work. The discussion should analyze first the topic on a general point of view, possibly stressing the key point that they tackle-context-attack-resolve through their work. If you need more indication, this is a smart reading  https://www.ncbi.nlm.nih.gov/pmc/articles/PMC4548568/

- sometimes the phrases are too long and difficult to understand. I suggest to re-edit the text (for example line 116-121 or 125-129 or 176-181).

Line 112: for the ethical sound, I would replace "interesting" with "challenging". 

Line 185: "the conclusions of this study still need to be further verified". What do the authors think that needs to be verified? The endoscopic management, the validation of ultrasound...? Please make the point clear. 

CONCLUSIONS

The section needs to be rewrite since contents are missing. The main topic disappeared.

I suggest to follow this scheme:

<<When writing your conclusion, you can consider the steps below to help you get started:

  1. Restate your research topic.

  2. Restate the thesis.

  3. Summarize the main points.

  4. State the significance or results.

  5. Conclude your thoughts.>>

(from: https://www.indeed.com/career-advice/career-development/how-to-write-a-conclusion-for-a-research-paper)

REFERENCES

The section does not need any improvement. 

I think your manuscript is a good work. 

I hope you will appreciate the suggestions. I remain at your disposal.

Best Regards

Author Response

Dear Reviewer:

Thank your comments concerning our manuscript "ID 1865825", titled "Clinical features and management of urethral foreign bodies in children:a 10-year retrospective study" .

Thank you so much for your help in our manuscript. Those comments are all valuable and very helpful for revising and improving our paper, as well as the important guiding significance to our researches. We hope that we can continue to discuss and communicate with you in addition to this manuscript in the future.

Generally speaking: We answered all comments one by one. We revised the manuscript according to suggestion of reviewers in main text, figures, tables and central image. We asked an English native language expert in the area of urology and four famous surgeon in the area to help as revised the manuscript. We changed some descriptions to make our manuscript more accurate and precise. We do hope these efforts can promote the quality of the manuscript.

ABSTRACT

  • Line 16: a space missing before "its treatment"

Answer: Thanks for your suggestion. We are so sorry for this, and we corrected this in the revised manuscript.

Change: we corrected this in the revised manuscript.

  • Line 19:I suggest to delete "eventually"

Answer: Thanks for your suggestion. We agree with you.

Change: we deleted "eventually" in the revised manuscript.

  • Line 20:the repetition of "most-mostly" makes the text rough. You can reformulate for example: the majority of boys was in puberal age and the main cause was the sexual gratification. 

Answer: Thanks for your suggestion. We agree with you.

Change: We have rewritten this sentence in the revised manuscript as "A majority of the boys(median age:11.8 years) were of puberal age, and the main cause of the UFBs was sexual gratification(94.1%). "(Line17-118, Page1)

  • Line 23:"and seasons (when grain is mature in rural areas)" is a deepening that does not allow an understanding of the issue. You may delete it or reformulate.

Answer: Thanks for your suggestion. We agree with you.

Change: We deleted this sentence from the revised manuscript.

  • Line 23-25:the phrase is too long and it is difficult to understand. Please improve also english syntax. 

Answer: Thanks for your suggestion. We feel so sorry for this, we asked an English native language expert in the area to help as revised the manuscript. We do hope these efforts can promote the quality of the manuscript.

Change: We have rewritten this sentence in the revised manuscript as"A majority of the boys(median age:11.8 years) were of puberal age, and the main cause of the UFBs was sexual gratification(94.1%). Girls were almost always in early childhood(median age:1.8 years), and most of the UFBs were related to specific clothing in specific regions and seasons. "

Point 1: you may specify if you refer to a rigid cystoscope or a flexible.

Answer: Thanks for your suggestion. We use a rigid cystoscope.

Change: We added "rigid" in the revised manuscript.(Line65, Page2)

INTRODUCTION

  • Line 36:"therefore, boy's UFBs often occur in adolescence" is a correct logic consequence of the previous phrase, but I suggest to add "as a consequence of self-plugging for sexual satisfaction", since this information is actually missing in the introduction. 

Answer: Thanks for your suggestion. We agree with you.

Change: We added "as a consequence of self-plugging for sexual satisfaction" in the revised manuscript.(Line31-32, Page1)

  • Line 40:"obvious clinical symptoms". I completely agree with you that related symptoms are obvious, but I suggest to specify them (since they are not in the entire introduction section) to allow the text understanding also to the readers who are not surgeons/urologists (according to Journal Scope). 

Answer: Thanks for your suggestion. Your suggestions are very valuable and will help readers to read this article more clearly.

Change: We added "(such as pain and hematuria)" after "obvious clinical symptoms" in the revised manuscript.(Line36, Page1)

MATERIALS AND METHODS

  • Point 2: in this section, as correctly stated in the abstract, authors should say "retrospective analysis was performed on patients who were admitted to our hospital due to UFBs from June, 2011 to June, 2021.". Add also this informations: a single-center and the name of the hospital center.

Answer: Thanks for your suggestion. We agree with you.

Change: We rewrote this sentence as "A single-centre retrospective analysis was performed on patients who were admitted to Children’s Hospital of Chongqing Medical University due to UFBs from June, 2011 to June, 2021." in the revised manuscript.(Line54-56, Page2)

  • Line 60:figure 1 contents are presented in the results section. I suggest to move it in the results section, since It shows some results.

Answer: Thanks for your suggestion. We agree with you.

Change: We moved this figure to the results section in the revised manuscript.

  • Line 62:please specify rigid or flexible cystoscope, which brand, degrees of optics and caliber.

Answer: Thanks for your suggestion. We agree with you.

Change: We added "(STORZ® , Fr9.5, 0°) " in the revised manuscript. (Line65, Page2)

  • Point 3: all the endoscopic procedures were performed under general anestesia? If you could resume this data you would add it. 

Answer: Thanks for your suggestion. All the endoscopic procedures were performed under general anestesia.

Change: We added "After general anesthesia " in the revised manuscript. (Line65, Page2)

  • Line 66:"indwelling catheterization for several time". It would be interesting if the authors specified the average time or standard deviation of catheterization in the results section. 

Answer: Thanks for your suggestion. We reviewed all the cases and collected and calculated this data.

Change: We added "The average indwelling duration of catheter was 3.5±1.3d and 0.5±1.0d for boys and girls, respectively." in the results section. (Line111-112, Page3)

  • Line 68: what do the authors intend for "review"? Is is a re-cystoscopy?

Answer: Thanks for your question. We're talking about outpatient visits, I'm sorry we didn't send the right message.

Change: We rewrote this sentence as "Children returned to the outpatient department for review once within 1 month after surgery" in the revised manuscript.(Line77-78, Page3)

RESULTS

  • I suggest to delete line 71-74.

Answer: Thank you for your suggestion, we missed that, and we deleted it in the revised manuscript.

Change: we deleted it in the revised manuscript.

  • Line 76: since the UFBs medium age is totally different in each gender (as the authors demonstrated), I suggest to offer a median age and SD calculated:

- for the male group

- for the female group

- for the entire sample of patients

The text may be rearranged as follows: a total of X patients wad admitted to our hospital, aged from .... to ...., with a mean age .... (SD ...). Boys were aged ...., mean .... . Girls were ...., mean..... . This construction, would enhance the size of the sample and would offer some prompt data for the eventual future use of your work.

Answer: Thank you for your detailed suggestion, the revision made the article more rigorous and clear. We appreciate your efforts.

Change: We rewrote this sentence as "A total of 40 patients wad admitted to our hospital(Figure 2), aged from 10 mo to 15 years with a mean age 6.6±5.1 years. Boys were aged from 4.5  to 15 years, mean age 11.7±2.5 years. Girls were aged from 10 mo  to 12.5 years, mean age 2.9±2.8 years(Figure 3). " in the revised manuscript.(Line85-88, Page3)

  • Line 77: "with or without"? Please specify how many of them had hematuria beyond the pain.

Answer: Thank you for your suggestion. We apologize for that, and we added this information to the revised manuscript.

Change: We added "All 17 boys presented for "pain(9 cases), pain with hematuria(4 cases), hematuria(3 cases), pain with infection(1 case)";" in the results section. (Line88-89, Page3)

  • Line 79: "the other children" is "the remaining girls"? 

Answer: We apologize for that,  "the other children" is "the remaining girls", we have corrected this phrase in the revised manuscript.

Change: We used "the remaining girls" instead of "the other children" in revising the manuscript.(Line91, Page3)

  • Line 79-80: "mainly because...." can be improved. For example you can say "since they were conducted to the hospital after their parents had found UFBs entering the urethral meatus that could not be removed". 

Answer: Thank you for your suggestion. We agree with you.

Change: We rewrote this sentence as "since they were conducted to the hospital after their parents had found UFBs entering the urethral meatus that could not be removed." in the revised manuscript.(Line91-93, Page3)

  • Line 87: a further conclusion would be "No simultaneous two or more different types of foreign bodies have been ever found".

Answer: Thank you for your suggestion. We agree with you.

Change: We added "No simultaneous two or more different types of foreign bodies have been ever found" in the revised manuscript. (Line96-97, Page3)

  • Line 88-95:since the diagnostic assessment usually precedes the management, this section might be replaced before "the foreign body types (line 81-82).

Answer: Thank you for your suggestion. Such modification makes the logic of the article more reasonable.

Change: We have changed the order of the two sentences in the revised manuscript.

  • Line 96:"stable" is "uneventful, regular"? Please, choose a more appropriate term.

Answer: Thank you for your suggestion. We apologize for that, in the revised manuscript, we have chosen a more appropriate word "satisfactory".

Change: we have chosen a more appropriate word "satisfactory" in the revised manuscript.

  • Line 96: "the follow-up period". How did they performed follow up? Uroflowmetry? Micturitction diary? Cystoscopy? This information is missing and needs to be specified in the section "materials and methods".

Answer: Thank you for your suggestion. We apologize for that, we should accurately describe the specific follow-up content, we added this part of content in the revised manuscript.

Change: we added "in addition to asking about clinical symptom relief, a routine urinalysis and ultrasonography were usually performed to evaluate the outcome of the surgery." to the section "materials and methods".

  • Line 97:an exceeding point at the end.

Answer: Thank you for your suggestion. We apologize for that.

Change: We corrected this in the revised manuscript.

FIGURES

  • Actually, for example, figure 3 is hardly readable. 

Answer: Thank you for your suggestion. We apologize for that, we should make the content of the figure more accurate to convey to the reader, rather than causing confusion to the reader. This figure shows the preoperative X-ray image of a child and the pictures of the foreign body removed after surgery.

Therefore, we have re-edited the contents of the image and the image annotation section. We hope that the added arrows and the image annotation section will help readers understand the image better.

Change: We have re-edited the contents of the image and the image annotation section in the revised manuscript.

DISCUSSION

  • sometimes the phrases are too long and difficult to understand. I suggest to re-edit the text (for example line 116-121 or 125-129 or 176-181).

Answer: Thank you for your suggestion. We apologize for that, We made a lot of mistakes in writing, which caused a lot of trouble to you and other reviewers, in the revision stage, we invited language experts to help us revise the whole manuscript, we do hope these efforts can promote the quality of the manuscript.

Change: We invited language experts to help us revise the whole manuscript.

  • Line 112:for the ethical sound, I would replace "interesting" with "challenging". 

Answer: Thank you for your suggestion. We agree with you.

Change:We replaced "interesting" with "challenging" in the revised manuscript.(Line137, Page7)

  • Line 185:"the conclusions of this study still need to be further verified". What do the authors think that needs to be verified? The endoscopic management, the validation of ultrasound...? Please make the point clear. 

Answer: Thanks for your question. We refer to the conclusions drawn in this study "Boys and girls have completely different ages and causes of onset. Ultrasound is a reliable method to diagnose urethral foreign body, and cystoscope is a reliable surgical method to treat UFBs".

Change:We added "Therefore, the conclusions of this study—that boys and girls have completely different ages and causes of onset, that ultrasound is a reliable method to diagnose UFBs, and that cystoscopy is a reliable surgical method to treat UFBs—still need to be further verified by other prospective studies with larger samples." in the revised manuscript.(Line212-215, Page9)

CONCLUSIONS

  • The section needs to be rewrite since contents are missing. The main topic disappeared.

Answer: Thank you for your suggestion. We apologize for that, we have rewritten this section according to the guidelines, and we hope that our corrections will be approved by you.

Change: We have rewritten the conclusion section as "The occurrence of UFBs in children is very rare, but it is challenging. Timely and effective management can improve outcomes for these children. Boys and girls have completely different clinical features. Ultrasound is a reliable method to diagnose UFBs, and cystoscopy is a reliable surgical method to treat them.

We hope that our study can provide diagnosis and treatment experiences for such rare diseases, improve the outcomes of such diseases, and improve treatments for these children" in the revised manuscript.(Line217-223, Page9)

Other answers:We are very grateful for your careful and rigorous review of the manuscript, which has greatly improved the quality of our manuscript. We know that it has cost you a great deal of effort. We would like to express our thanks again and look forward to further learning from you in the future.

Best Regards,

Xiangpan Kong, M.Med.

Reviewer 4 Report

No comments

Author Response

Dear Reviewer:

Thank your comments concerning our manuscript "ID 1865825", titled "Clinical features and management of urethral foreign bodies in children:a 10-year retrospective study" .

Thank you for your valuable time in reviewing our manuscript, and thank you for your high recognition of our manuscript. It is an honor to get your support and thank you again for your efforts.

Best Regards,

Xiangpan Kong, M.Med.

Round 2

Reviewer 1 Report

I would like to congratulate the authors on their work. In the revised manuscript, the authors have addressed all my comments. The overall scientific quality of the manuscript has improved significantly.

Author Response

Dear Reviewer:

We are so grateful for your careful and rigorous review of the manuscript, which has greatly improved the scientific quality of our manuscript.

Best Regards,

Xiangpan Kong, M.Med.

Reviewer 2 Report

Dear authors,

I ask that you read your own work in detail once more and rewrite it, because there are still a number of shortcomings and inconsistencies in the article.

In addition to adding references 7, 8, and 9, a textual explanation should be added for them in the manuscript itself, as written in the answer to me.

Line 73, line 375 - perhaps it would be better to write conditions instead of diseases

Line 108 – no formal analysis?

In Table 1, you must write the meaning of the abbreviation ND.

The numbers in Figure 2 still do not match the numbers in the text! Please re-read your own work in detail and make the necessary changes!

In Table 1, four boys were subjected to an open surgical method, while in Figure 2, 3 were mentioned.

In figure 3, in addition to percentages, absolute numbers must also be written. Put the percentages in brackets.

Line 231 - network induction? Do you mean influence?

Line 361 - However, a total of 56 beads were removed intraoperatively (Figure 5).?

Etc….

Author Response

Dear Reviewer:

Thank your comments concerning our manuscript "ID 1865825", titled "Clinical features and management of urethral foreign bodies in children:a 10-year retrospective study" .

Thank you so much for the great effort you have put into our manuscript. We have carefully taken your comments into consideration in preparing our revision, which has resulted in a paper that is rigorous and more compelling. Revised portion are marked red in the paper. The corrections in the paper and the responds to the comments are as flowing:

Comment 1: In addition to adding references 7, 8, and 9, a textual explanation should be added for them in the manuscript itself, as written in the answer to me.

Answer 1: We apologize for that. We added these contents in the revised manuscript.

Change 1: We added "Decide whether to use antibiotics based on whether the child has a urinary tract infection, In children who require intravenuous treatment tobramycin or gentamicin was used if there is normal kidney function. When abnormal kidney function was suspected, ceftriaxon or cefotaxime were alternative treatment options. In children who can receive oral treatment without any known resistant urinary cultures, cefixime or amoxicillin-clavulanate are the empirical treatment options. The selection of antibiotics were flexible according to the actual situation[7]. whether to catheterize or not and the duration of catheter indwelling should be related to the degree of urethral injury and the presence or absence of UTI, For children with little or no urethral injury, no catheterization or temporary catheterization after surgery is safe, for those who have injured the urethra with a foreign object, a short catheterization (within 3 days) can reduce bleeding and relieve pain, finally, for those children with definite or high suspicion of complicated UTI, catheterization for a week or so is necessary[8-9]." in Line73-86, Page3.

Comment 2:Line 73, line 375 - perhaps it would be better to write conditions instead of diseases

Answer 2: Thank you for your suggestion. We agree with you.

Change 2: In the revised manuscript, we used "conditions" instead of "diseases" in Line50, Page2, Line146, Page8 and Line229, Page10.

Comment 3: Line 108 – no formal analysis?

Answer 3: We apologize for that. We rewrote the text to include formal analysis.

Change 3: We rewrite this text as "Preoperative X-ray examination was performed on all boys to determine the location and size of the foreign bodies. All 23 girls underwent preoperative ultrasound. Among them, the ultrasound report of 12 girls showed negative, and one of these 12 girls did not undergo surgery because their parents refused to do so, the surgical results of the remaining 11 girls showed negative, and no foreign body was found, which was consistent with the ultrasound results. The other 11 girls with positive ultrasound findings received surgical treatment, and 9 cases were found and successfully removed foreign bodies, while the other 2 cases were not found foreign bodies. It is worth noting that the parents of 1 of these 2 children clearly reported a history of self-removal of foreign bodies after preoperative ultrasound examination" in Line112-122, Page3, 4.

Comment 4: In Table 1, you must write the meaning of the abbreviation ND.

Answer 4: We apologize for that. We have added it in the revised manuscript.

Change 4: We added "·ND: Not found, M: male, F: female" in Line128, Page5.

Comment 5: The numbers in Figure 2 still do not match the numbers in the text! Please re-read your own work in detail and make the necessary changes!

In Table 1, four boys were subjected to an open surgical method, while in Figure 2, 3 were mentioned.

Answer 5: We apologize for that. Repeated revisions of the original manuscript caused some errors in the data, for which we are deeply sorry. Therefore, according to your comments, we read and checked our manuscript repeatedly and corrected all errors in the data. Table 1 is our original data, it is correct, Figure 2 has a series of errors, we have corrected these errors and redrawn the figure.

Change 5: We corrected these errors, redrawn the figure. A new Figure 2 has been re-uploaded.

Comment 6:In figure 3, in addition to percentages, absolute numbers must also be written. Put the percentages in brackets.

Answer 6: Thank you for your suggestion. We agree with you.

Change 6: We've added the exact numbers. And we can't put the percentages in brackets because the drawing software doesn't support it.

Comment 7: Line 231 - network induction? Do you mean influence?

Answer 7: Thank you for your suggestion. We agree with you.

Change 7: In the revised manuscript, we used "influences" instead of "induction" in Line154, Page8.

Comment 8: Line 361 - However, a total of 56 beads were removed intraoperatively (Figure 5).?

Answer 8: We apologize for that. We went over the entire manuscript and corrected these errors in the data.

Change 8: We corrected these errors, the correct number was "52", which we revised.

Other Answers: It is our great honor to get your professional and rigorous review comments, we have made many mistakes when writing original articles, with the help of your and other referees, we not only revised the mistakes, but also improved the scientific quality of the article. We know that it has cost you a great deal of effort.

We would like to express our thanks again, and we wish that we can continue to discuss and communicate with you in addition to this manuscript in the future.

Best Regards,

Xiangpan Kong, M.Med.

Round 3

Reviewer 2 Report

Dear authors,

You probably meant formal analysis instead of no formal analysis (line 92).

Please correct the English language by a native speaker.

Author Response

Dear Reviewer:

Thank your comments concerning our manuscript "ID 1865825", titled "Clinical features and management of urethral foreign bodies in children:a 10-year retrospective study" .

Thank you so much for the great effort you have put into our manuscript. We have carefully taken your comments into consideration in preparing our revision, which has resulted in a paper that is rigorous and more compelling. Revised portion are marked red in the paper. The corrections in the paper and the responds to the comments are as flowing:

Comment 1: You probably meant formal analysis instead of no formal analysis (line 92).

Answer 1: Thank you for your suggestion. We agree with you. In fact, a formal analysis did occur in the revised manuscript. We neglected to modify this statement.

Change 1: In the revised manuscript, we replaced "no formal analysis" with "formal analysis" .(Line 92, Page 3)

Comment 2: Please correct the English language by a native speaker.

Answer 2: We apologize for that. In fact, in the first revision of the manuscript, we have asked a native language expert to help us edit the language. Since we revised part of the content in the second revision, we invited a native language expert to edit the language again, hoping to improve the language quality of the manuscript.

Change 2: In the revised manuscript, We have invited an English native language expert to check the revised version of our paper to ensure correctness of the spelling, grammar and syntax again.

Other Answers: It is a great honor for us to get your professional and patient guidance to complete this research.

We hope to learn such spirit from you in our future work and study. It has been a very valuable learning experience. Thank you again for your great contribution.

Best Regards,

Xiangpan Kong, M.Med.
